# An Analysis of Mushroom Consumption in Hungary in the International Context

**Bernadett Bringye** [1], **Maria Fekete-Farkas** [2] and **Szergej Vinogradov** [2,*]

[1] Institute of Business Regulation and Information Management, Hungarian University of Agriculture and Life Sciences, Páter K. u. 1, 2100 Gödöllő, Hungary; Almadi.Bernadett@uni-mate.hu

[2] Institute of Economic Sciences, Hungarian University of Agriculture and Life Sciences, Páter K. u. 1, 2100 Gödöllő, Hungary; Farkasne.Fekete.Maria@uni-mate.hu

[*] Correspondence: Vinogradov.Szergej@uni-mate.hu

**Abstract:** It is hardly an exaggeration to state that producing and consuming mushrooms may provide an answer to several of the challenges facing mankind. This research is related to the UN sustainable development goals relative to different issues. First of all, mushroom production uses agricultural and industrial byproducts as inputs and being labor intensive contributes to the job and income creation for undereducated people in less developed areas. In addition, as mushrooms have high protein content and they are a suitable alternative for meat for populations with a diet lacking in variety; at the same time, they also have the potential for food connoisseurs and consumers who make conscious and educated choices to improve their diet by using healthful and environmentally friendly methods. The nutritional value of mushrooms means that consumption could be an important supplementary therapy for several illnesses. The key issue of sector development is the increasing demand. In order to address this, investigation and research related to consumer behavior is needed. The aim of this research was to explore the dimensions of Hungarian mushroom consumer behavior and to segment Hungarian consumers. An online questionnaire survey was conducted between December 2019 and February 2020 and the final sample of 1768 respondents was considered for the purposes of analysis. Exploratory factor analysis was used to identify groups of correlating variables describing mushroom consumption. The authors identified four dimensions of Hungarian mushroom consumer behavior: (1) medicinal and functional properties, (2) consumption for enjoyment, (3) supplementary food source, and (4) negative assessment of the product range. Using cluster analysis, three consumer groups were identified: (1) health-conscious consumers, (2) indifferent consumers, and (3) average consumers. The research results indicated that consumers' sociodemographic characteristics (age, educational level, marital status, and place of residence) have a significant impact on mushroom consumption behavior. The results of this paper can have implications for policy makers and business management in diversifying their production and selecting marketing tools.

**Keywords:** sustainability; mushroom production and consumption; consumer behavior; consumer segmentation; diet and health issues; functional food; SEM





## 1. Introduction

The United Nations' Agenda 2030 (Transforming our world: The 2030 Agenda for Sustainable Development) puts forward 17 aims for the future and several are related to the potential benefits of the production and consumption of mushrooms. However, relatively few publications mention these issues.

For instance, Target 2 described in Agenda 2030 entitled Zero Hunger by 2030 aims not only to eliminate hunger and undernutrition but also all forms of malnutrition, i.e., unbalanced diets or inadequate intake of nutrients. Diets need to be transformed and entire populations need to be fed with food sources that are relatively cheap, cost-effective, and have low environmental impact [1,2].

Another example is Target 12, which is Sustainable Consumption and Production, which requires not only the complete transformation of the use of natural resources, production technology, and consumer behavior but also the elimination of food waste at all stages of the food marketing chain. For a large share of the world population, meat is the most important source of protein in their diets, however, the increasing consumption of meat may have negative effects on health and sustainability and has a large contribution to greenhouse gas emissions and climate change [3–5]. There are increasing efforts for finding and using alternative sources of protein, for instance, encouraging people to consume more fish and pulses or some really novel and less accepted alternatives, such as insects and seaweed. However, some traditional foods, for example, mushrooms, are acquiring less attention. There is a research gap that exists: The advantages of mushroom consumption are worth investigating, with a special focus on what factors motivate its consumption, how the consumer preferences can be changed, and what kind of public action is also needed for increasing the profitability of mushroom production.

Target 8 of UN Sustainable development strategies is as follows: Promote sustained, inclusive, and sustainable economic growth. Full and productive employment and decent work for all could benefit significantly from an increase in the production and processing of mushrooms: due to its labor intensity, this sector provides income for small and medium-sized enterprises and for undereducated rural populations [6].

Target 13 includes Climate actions and is also in line with the characteristics of mushroom production since the sector relies heavily on waste products generated by other branches of agriculture, horticulture, and forestry, contributing to recycling and circular economy. Mushrooms can produce organic waste and byproducts and the organic waste generated by its supply chain can be used for soil improvement. One of the largest contributors of $CO_2$ emission is the transportation sector. Normally, mushrooms have short food supply chains because major input is found locally and a large share of the products is sold or processed within a short distance. Mushrooms deteriorate rapidly and thus the number of intermediaries tends to be small. According to the American Mushroom Institute, mushrooms have a big advantage in their contribution to $CO_2$ emission as well. In comparison with other protein foods consumed, mushrooms have only 0.5 kg of $CO_2$ per pound, whereas chickens have 3.1 kg $CO_2$ per pound, and pork has 5.5 kg $CO_2$ per pound. Cheese is at 6.1 kg $CO_2$ per pound and eggs are 2.2 kg $CO_2$ per pound consumed [7–9].

Target 9 of SDGs is to ensure healthy lives and promote well-being for all individuals at all ages, which is extremely relevant in this age of COVID-19 as mushrooms contain valuable minerals and vitamins in large quantities that contributes to the strengthening of the immune system. It is widely used in medicine in many fields and in the beauty industry. Forest mushroom gathering with appropriate expertise is an excellent leisure program [7,9,10].

Despite the favorable climatic conditions in Hungary, mushroom production is low and thus its contribution to solving environmental problems, job creation for the low-skilled rural population, generating income for small and medium-sized enterprises, and improving the quality of life, including nutritional and other issues, is fairly low as well. The basic assumption of our research is that the main factor in the solution is the increase in consumption. In order for appropriate public programs to be developed for this and for companies to be able to develop an appropriate production and marketing strategy, it is necessary to know current and potential consumers and their mushroom-related knowledge while paying special attention to the criteria for sustainable development. The second half of the 20th century was characterized by a dramatic rise in the quality of life, resulting in an increase in the consumption of meat products and a decrease in fruits and vegetables; the latter is still lagging far behind WHO recommendations [11]. Healthy lifestyle has started to make an appearance in the past decades as a result of political and marketing campaigns; however, change is still extremely slow. According to Pfau et al. [12], mushroom consumption in Hungary is peaking at 1.5 kg per person per year, which lags behind the international average of 4 kg per year [13].

This paper is organized according to the following structure. This part of the Introduction Section is followed by an extensive literature review of historical and geographical views of mushroom consumption and production focusing on sustainability issues; Section 2 presents the research methodology, including presentation of data collection methodology and analysis procedures are also discussed. Section 3 includes relevant research findings followed by a discussion in Section 4, Section 5 highlights the main conclusions of the study and emphasizes its potential political and managerial implications. In the final section, the limitation of research and further research areas are presented.

*1.1. The Significance of Mushroom Production and Consumption in a Historical and Geographical View*

The collection of edible mushrooms is as old as humankind itself; mushrooms were likely collected alongside plants by early humans. According to [14], the extensive production of shiitake mushrooms using wooden logs started in the tropical regions of Asia several millennia ago. Both the Quran and the Bible refer to the consumption of mushrooms; according to the Bible, truffles have heavenly origins [15]. This species was considered a royal or divine delicacy in ancient times and Aristotle (384 B.C.) considered it "the food of the gods". Claudius, an Emperor of Rome (10 B.C.), professed that "a plateful of scaly wood mushrooms is worth more than any triumph on the battlefield". In Egypt, mushrooms were a gift from Osiris and had a significant role in the pharaohs' diets [16].

Mushrooms have also been used for medicinal purposes. During the Han dynasty in China (206 B.C.–220 A.D.) reishi mushrooms, called ling chi in Chinese, were considered as an ingredient of immortality [17]. In 1st century ancient Rome, Gaius Plinius utilized mushrooms to treat spider and scorpion bites as well as to heal pulmonary diseases [18]. The first known descriptions of mushrooms also date back to China, 1245 A.D.; Chen Yen Yu refers to a manuscript by Teofrastos from ancient Greece [14,19].

Lelley [20] refers to the manual Le Jardinier Français (The French Gardener), by De Bonnefons in 1650, as the beginnings for farming button mushrooms. In 1676, Cardilucius considered mushrooms to be a major antidote to leprosy [21]. Charles de L'Escluse (1526–1609) is widely regarded as the founder of mycology and his publication "Fungarium in Pannoniis observatorum brevis historica" (1601) is the first known document describing mushrooms in Hungary [22].

At this time period, button mushrooms were already being sold in the markets around France; however, commercial production only started in the mid-19th century. Lonicerrus Amamus described wood ear mushrooms and giant puffballs in 1679 [19]. French botanist Teurnefort published a volume on mushrooms in 1707 [23]. Carl von Linné laid the foundations of modern mycology and described several species as well in 1753 [23]. Truffles growing in the wild were a major export product from the historical territory of Hungary in the 19th century; during the same time period, there were early attempts in Wien to produce them commercially [24].

Mushrooms are a healthful and tasty source of nutrition for humankind; however, they were not widely consumed due to the fact that up until the mid-19th century, their farming was at a basic level. Mushroom production requires advanced knowledge and technical skills; in addition, the produce needs to be consumed or processed within a short period of time.

Important technological advances were made by Duggar in 1905 who describedspawn making [14]; by Lambert in 1918 by improving spawn production; and by Sinden in 1932 who was the first to produce grain spawn [25]. China was the first to farm shiitake mushrooms using extensive methods (1000–1100 A.D.); whereas the farming of oyster mushrooms started in the USA at the beginning of the 20th century and button mushrooms in 17th century France [26]. Hungary was considered a major mushroom- producing country in 1938, ranking third in the world (behind France and the USA) with an annual output of 600 tons and 200,000 m$^2$ farming area. Mushroom spawns from Hungary were high quality and widely sought after. Naturally, WW2 put an end to production and nationalization in 1948 devastated the mushroom sector. Production only started to increase

again during the 1990s [22]. Currently, mushroom production and consumption worldwide demonstrate steady growth, which is still inadequate when potential development is considered. China, the USA, The Netherlands, India, and Vietnam have increased at the greatest rates [27]. According to Ikar [28], Russia is also joining major global players: export tripled within a single year from 2019 to 2020. Currently, China and the USA produce the most mushrooms in the world [13], whereas, in Europe, the market leaders are Poland and The Netherlands. Twelve species dominate the market of farmed mushrooms:

- Button mushrooms (*Agaricus bisporus* and *bitorquis*);
- Shiitake (*Lentinula edodes*);
- Oyster mushrooms (*Pleurotus* sp.);
- Enoki (*Flammulina velutipes*);
- Wood ear mushrooms (*Auricularia*, especially *Auricularia judea*);
- Shaggy ink cap (*Coprinus comatus*);
- Straw mushrooms (*Volvariella* sp.);
- Ram's head (*Grifola frondosa*);
- Nameko (*Pholiota nameko*);
- Reishi (*Ganoderma lucidum*) [29].

However, only the first five of this list are farmed and produced in significant amounts [30]. Ethnomycologists who explore the relationship of various cultures to mushrooms divide nations to mycophiles (those who favor mushrooms) and mycophobes (those who reject or fear mushrooms). Poland, Russia, China, and France are examples of the former, whereas England, Canada, the USA, Australia, and New Zealand belong to the latter category.

There has been a significant increase in recent years in the demand for mushrooms for medicinal purposes; for instance, the global trade in reishi mushrooms is approximately USD 2 billion annually [31].

Currently, mushroom production in Hungary demonstrates gradual increase, with the following distribution of species:

- Button mushrooms (90–91% of all produced quantity);
- Oyster mushrooms (7–8%);
- Exotic mushrooms (shiitake, reishi, shaggy ink cap, and sheathed woodtuft at 1–2%) [32]. Around 75% of button mushrooms are sold fresh and 25% in processed form.

### 1.2. Impontance of Mushroom Consumption Focusing on Nutrient and Health Effects

"Let food be thy medicine, and medicine be thy food"—consumers are increasingly taking Hippocrates's 2500 year old advice to heart these days [33]. Due to the rising costs of health care, higher life expectancy, and higher quality of life for the older generations, functional foods are on the rise [34].

In recent years, diets have started to become more conscious and scientifically based; several trends and counter-trends have emerged in popular culture as well. Flavors, fragrances, and textures are naturally still the main considerations; however, more emphasis is placed on bioactive content of various foods [35]. As a consequence, several products containing mushrooms have entered the markets in the Far East and in Europe, mainly in the form of medicinal products. Mushrooms are making an appearance worldwide as ingredients in coffee, chocolate and tea, as well as in cosmetics.

According to Geösel et al. [36], there is a steady increase in interest in natural products with possible positive health effects and medicinal properties. For instance, there is a high demand in almond-scented powdered button mushroom capsules, which is difficult to meet using the current extensive technology.

In conclusion, it can be stated that with the current spread of healthy lifestyles, vegan and vegetarian diets, and an increasing interest in organic produce, mushroom consumption is on the rise worldwide.

According to Royse [30], worldwide consumption of mushrooms increased from 1 kg per person per year to 4 kg within a very short time period. Naturally, mushroom

consumption varies widely across countries. Europe is the largest market for mushrooms, comprising over 35% of global markets. Demand is on the rise in North America and even more sharply expanding in South America. In the meantime, Africa and the Middle East exhibit a moderate expansion [32].

Some Asian countries have mushrooms as a central part of their cuisines, mostly oyster mushrooms and local species; they have been aware of the health benefits of consumption for a long period of time [27].

Button mushrooms are mostly consumed in Western Europe [31]. Many African countries rely on mushrooms collected in the wild in the rainy season as part of their traditional diets. Farmed mushrooms have made an appearance but they are lagging behind other continents.

In Hungary, the consumption of mushrooms varies between age groups and other demographic characteristics, such as educational background.

Mushrooms were often considered meat substitutes already at the beginning of the 20th century [37], even though their protein content is merely 2–5%. Proteins sourced from mushrooms are considered nearly equivalent to those from animal sources [38]. Recently, it is viewed as an alternative protein source [39]. Mushrooms have a high mineral and vitamin content and they are a source of valuable amino acids and fatty acids as well as natural vitamin D [40]. They have a low glucose content and low calories; however, they are high in natural fibers and thus they come highly recommended for those requiring special diets, such as diabetics [18]. Finimundi et al. [41] studied the positive effects of shiitake mushrooms in cancer patients and its relation to lentinan and emitanin 1 poly-saccharide content [38].

Several mushroom species have proven positive effects on the immune system. Mallard et al. [42] investigated the concurrent use of reishi, shiitake, and ram's head mushrooms and concluded that their cumulative positive effect is more potent.

Table 1 summarizes the medicinal uses of mushrooms by disease groups. It is worth mentioning that these diseases are the leading cause of death across Europe and even Hungary.

**Table 1.** The significance of medicinal mushrooms.

| Medicinal Effect | References |
|---|---|
| Tumors | Finimundy et al. (2018) [41]; Cerletti et al. (2021) [43] |
| Immune system | Mallard et al. (2019) [42]; Villares et al. (2012) [44] |
| Cardiovascular diseases | Lelley (2018) [20]; Rahman et al. (2015) [45], Genesan–Xu (2018) [46] |
| Inflammations | Berg–Lelley (2016) [47] |
| Digestion problems | Shang et al. (2013) [48]; Wong et al. (2013) [49]; Wang et al. (2018) [50]; Kumari (2020) [51] |
| Autoimmune deficiencies | Beelman et al. (2019) [52]; Muszyńska et al. (2018) [53] |
| Antibacterial and antiviral effects | Vetter (2010) [18] |
| Reducing blood glucose level | Calvo et al. 2016 [54]; Vitak et al. (2017) [55] |
| Skin care | Wu et al. 2016 [56]; Usman et al. (2021) [57] |

### 1.3. Policy Actions and Public Campaign for Changing Consumer Behavior

Several national and commercial campaigns were organized worldwide to promote mushrooms. Certain countries make mushroom farming a national priority [58].

Following in the footsteps of mushroom projects in the USA [59] and Australia [60], a "School Mushroom" project initiated by Biofungi Ltd. has been an excellent project introducing the basics of mushroom production to schoolchildren [61]. As a response to the economic fallout of the COVID-19 pandemic, a national campaign urged consumers to purchase locally grown produce in order to protect local jobs [62].

*1.4. Impotance of Mushroom Production Focusing on Environmental Benefit and Contribution to the Rural Development*

Climate change is having a negative effect on food production; therefore, we need to increase our consciousness in production as well as in consumption. It is imperative to decrease the reliance on animal proteins, over-consumption of calories, and focus on programs promoting healthful diets [63]. This is where the production and consumption of mushrooms enters the picture. The production of mushrooms is different from any branches of plant cultivation or horticulture since its environmental footprint is exceptionally low. According to Monterey Mushroom [64], the production of one pound of mushrooms requires 1.8 L of water and 1.0 kWh of energy; in addition, it relies on by-products from the horticultural, forestry, animal husbandry, and food processing sectors to generate food with high biological properties [65,66]. Moreover, mushroom farming is also a method to improve soil quality [67] and to produce raw materials for packaging that substitutes plastics [68].

In the case of certain Asian and Eastern European countries, mushroom collection from the wild is often not only a source of nutrition but also a source of income for impoverished populations. [69–71].

According to Chang [72], humanity has been facing three major challenges since the turn of the century and mushroom production and consumption could be a step in the right direction, namely: the challenges of pollution, lack of adequate food supplies, and degradation of the quality of life. Table 2 below summarizes the global challenges and research areas related to mushroom production and consumption.

**Table 2.** A summary of related literature: The significance of mushroom production and consumption in solving global problems.

| Global Challenges | Functions | Major Publications | Underlying Concept |
|---|---|---|---|
| POLLUTION | Waste management | Gyenge et al. (2016) [65]; Gunady et al. (2012) [73]; Cunha Zied et al. (2020) [74]; Kumla et al. (2020) [75] | Mushroom production uses waste and by-products from various agricultural sectors. |
| | Soil improvement | Klein 2020 [67] | Mushrooms accumulate toxins from the soil. |
| LACK OF FOOD SUPPLIES | Food production | Royse et al. (2017) [13] | Mushrooms are healthful and safe sources of nutrition (low carbohydrate content, high protein and fibre content). |
| DEGRADATION OF THE QUALITY OF LIFE | Reduction in poverty | Imtiaj−Rahman (2008) [69]; Barmon et al. (2012) [70]; Bajpai et al. (2021) [71] | Mushroom production and collection of forest mushrooms provide regular jobs and income for populations with low levels of education. |
| | Mycotherapy | Zhang et al. (2018) [76]; Glamoclija−Sokovic (2017) [77] | Mushrooms are used for medicinal purposes. |

*1.5. Research Model and Hypotheses Development*

The hypothesis system of the research is related to the exploration of certain dimensions of mushroom consumption habits, as well as to the examination of the correlations between these dimensions. Table 3 summarizes the descriptions of the research dimensions (factors) and the literature sources related to the factors.

**Table 3.** Literature sources and empirical studies related to the research dimensions.

| Dimensions (Factors and Constructs) | Short Description of Indicators | Literature Sources Related to Factors |
|---|---|---|
| Medical and functional properties | Consumption due to medical and functional reasons | Pender (1987) [78]; Wang et al. (2020) [79]; Papp-Bata−Szakály (2020) [80]; Ronteltap (2008, p. 91) [81] |
| Consumption of enjoyment | Consumption for tastes, aromas, special culinary delights | Steenkamp (1997, p. 144) [82]; Thomson−Crocker (2015, pp. 343–353) [83] |
| Supplementary food source | Consumption as a meat substitute or as ingredients for special meals | Pilgrim (1957, pp. 171–175) [84] |
| Negative assessment of the mushroom product range | Negative assessment of the choice of mushrooms and mushroom products | Almádi (2021) [85] |

Based on the findings of the studies on mushroom consumer attitudes [80,86], the functions of mushrooms (medical, functional, enjoyment, and food supplement) are not markedly different from each other and thus the authors set up the first hypothesis as follows.

**Hypothesis 1 (H1).** *The dimensions related to the mushroom functions are in a positive relationship with one another.*

Almádi [85] has shown that with the increase in mushroom knowledge, European consumers are less satisfied with the range of mushroom products. Hence, the authors proposed the second hypothesis.

**Hypothesis 2 (H2).** *A negative assessment of the mushroom product range is positively related to the dimensions associated with the mushroom functions.*

Several studies related to food consumption habits [87–91] have demonstrated that socio-demographic characteristics significantly influence food consumption patterns. The authors therefore formulated the third hypothesis as follows.

**Hypothesis 3 (H3).** *Social and demographic factors (age, marital status, educational level, and location) have a significant impact on mushroom consumer behavior in Hungary.*

## 2. Materials and Methods

This chapter presents the detailed research framework and steps of the research which can be divided into two parts. The research was focused on Hungarian adults' consumer habits and, at the same time, we examined the attitudes of university students towards mushrooms (Figure 1).

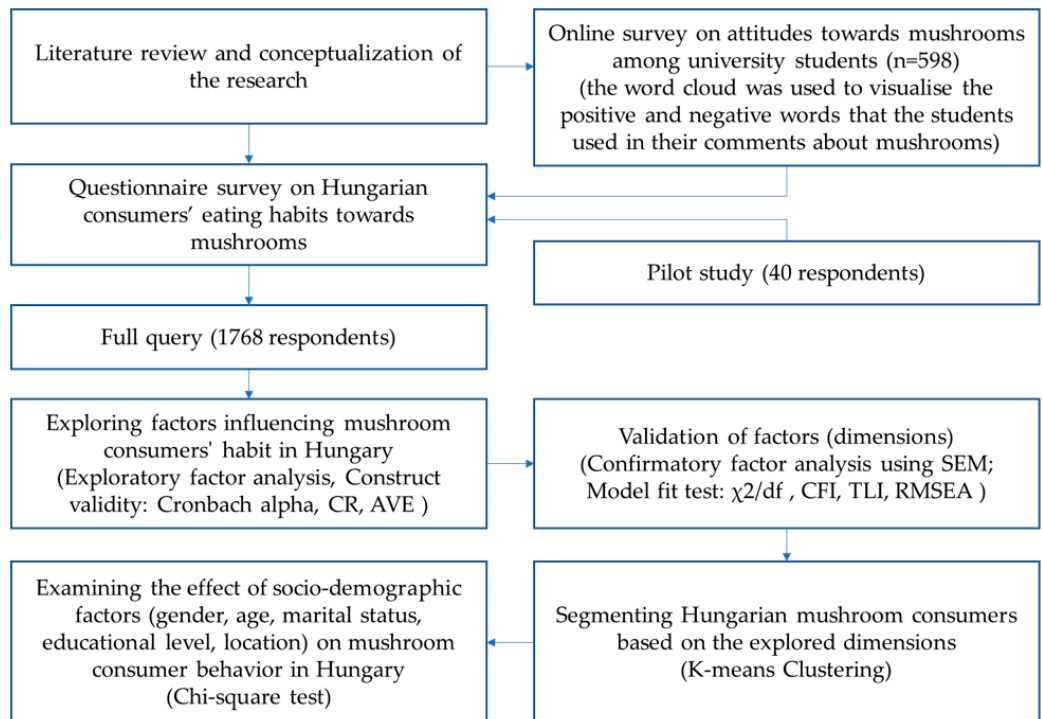

**Figure 1.** Research framework for examining factors influencing consumer behavior towards mushrooms in Hungary.

During the literature review and the development of the research concept, attitudes towards mushrooms were explored among university students. Respondents had to answer a simple question: "What first comes to mind when you hear the word mushroom?" A sample of 598 respondents was achieved. The word cloud was used to visualize the positive and negative words that the students used in their comments about mushrooms.

In order to survey consumer habits in Hungary, the next step consisted of an online questionnaire administered using LimeSurvey open-source survey software. Given that it was not possible to obtain an appropriate sampling frame consisted of consumers, convenience sampling was adopted, which had been used for studying mushroom consumption and purchasing behavior. As there are no variables that can be measured directly, mushroom consumer habits are difficult to survey or quantify. Based on the dimensions identified in the research, the statements used to examine each area are summarized into latent variables.

Items for measuring latent variables (constructs) were adapted from Pender's health support model [78], Wang's Healthy Dietary Behavioral Model [79], Steenkamp's food consumer model [82], and Ajzen and Fishbein's model of designed behavior [92]. In the pilot phase, the questionnaire was first tested using 40 respondents. Subsequently, 2017 respondents filled in the questionnaire, out of which 1768 were complete and used for analysis.

Participation was voluntary and subjects were chosen randomly and anonymously. Respondents were able to rate the items using 7-point Likert scales (1 = strongly disagree, 7 = strongly agree).

The questionnaire included the following sections:

1.  General questions about mushroom consumption: the frequency and structure of mushroom consumption, the choice of mushroom products in Hungary, and changes in consumption habits;
2.  Health effects of mushroom consumption: awareness of the health effects of mushrooms;
3.  The respondents' eating habits and socio-demographic characteristics.

Regarding the demographic profiles of respondents, 58% of them were female and 42% were male. Table 4 summarizes distribution by age groups that are thought to be relevant since attitudes and consumer habits and motivations may vary.

**Table 4.** Socio-demographic characteristics of the respondents.

| Variable | Category | Frequency | Percentage |
|---|---|---|---|
| Gender | Male | 749 | 42.0 |
| | Female | 1036 | 58.0 |
| Marital status | Single | 706 | 39.6 |
| | Married | 946 | 53.0 |
| | Divorced/widowed | 133 | 7.5 |
| Age | Under 20 | 193 | 10.8 |
| | 20–30 years of age | 661 | 37.0 |
| | 31–40 years of age | 247 | 13.8 |
| | 41–50 years of age | 368 | 20.6 |
| | 51–60 years of age | 177 | 9.9 |
| | 61–70 years of age | 100 | 5.6 |
| | Over 70 | 39 | 2.2 |
| Location | Capital | 568 | 31.8 |
| | City | 221 | 12.4 |
| | Town | 632 | 35.4 |
| | Village | 180 | 10.1 |
| | Small village | 184 | 10.3 |
| Education | Primary school | 77 | 4.3 |
| | Vocational school | 153 | 8.6 |
| | Secondary school | 913 | 51.1 |
| | College/university | 642 | 36.0 |

Exploratory factor analysis (EFA) was performed on the first half of the sample chosen at random in order to assess the construct validity of the scales evaluating consumer behavior towards mushrooms. Harman's single factor test [93,94] and common latent factor (CLF) [95,96] were used to examine potential common method variance (bias due to using a single data collection method). The EFA results indicated that the single factor explained 24.30% of the variance in the items, indicating that common method bias was unlikely to be a problem in the model. Differences between the standardized regression weights in the model with the CLF with those in the model without the CLF were all less than 0.2, suggesting that common method biases did not affect the findings.

The structure suggested by EFA was subsequently validated by carrying out Confirmatory factor analysis (CFA) on the second half. Statistical analyses were performed using IBM Statistics SPSS Version 25 and AMOS Graphics Version 23.0.

The reliability of the latent structures was confirmed using the following indicators:

— Cronbach alpha coefficient: Values over 0.6 indicate a reliable latent variable [97];
— Spearman–Brown coefficient: Values over 0.6 are acceptable [98];
— Average variance extracted (AVE): Values over 0.5 are acceptable [99,100];
— Composite reliability (CR): It expresses the shared variance for the latent variables that comprise the observed indicators. According to Hair et al. [99], all the latent variables in the model need to have a CR of at least 0.7 each.

In cases when average variance extracted is below 0.5 but composite reliability exceeds it, the reliability of the latent variable is acceptable [101]. After the validation of the measuring model, the next step was the CFA using structural equations modeling (SEM). Model fit was considered acceptable if $\chi^2/df \leq 5$ [102,103], CFI, and TLI values were >0.90 [100] and RMSEA <0.08 with the 90% CI upper limit <0.08 [104].

Segmenting of Hungarian mushroom consumers was conducted using K-means clustering, since this is the recommended procedure for larger samples (instead of hierarchical clustering).

Chi-square test was used to examine the effect of socio-demographic factors (gender, age, marital status, educational level, and location) on Hungarian consumers' eating habits towards mushrooms and mushroom products.

### 3. Results

*3.1. Research of Associations Using the Word Cloud Method*

Parallel with the main research direction, the authors attempted to explore the extent of positive vs. negative attitudes toward mushrooms by Hungarian consumers. It was considered to be an ideal method to reveal enthusiasm as well as aversion. During the process of word cloud creation, adjectives were sorted into groups of positive, negative, or neutral.

Out of the 598 responses provided, 256 (42.8%) were assessed as positive, 93 (15.5%) as negative, and 249 (41.6%) as neutral. Positive adjectives were mainly related to taste and emotions and sometimes memories. Negative responses were usually related to texture, flavor, poisoning, or its treatment. Dishes containing mushrooms as ingredients or names of mushroom species mentioned in responses were categorized as neutral.

The questionnaire prompt was to say the first association that comes to mind regarding mushrooms. Hardly any responses referred to the collection of mushrooms in the wild and most were related to farmed mushrooms. The results are illustrated in Figure 2.

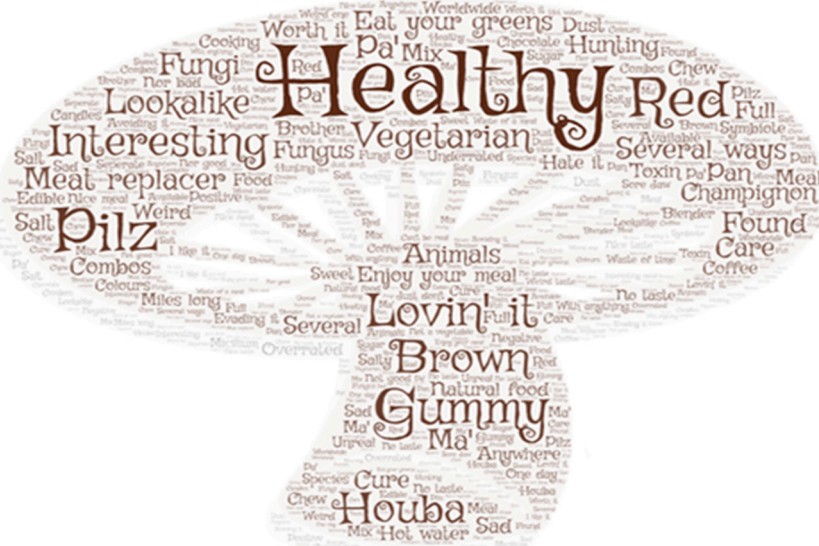

**Figure 2.** Word cloud of word associations related to mushrooms (source: own research, N = 598).

*3.2. Exploring Dimensions of Hungarian Mushroom Consumer Behavior*

Using Exploratory factor analysis (EFA), four dimensions related to mushroom consumption were identified (Table 5):

1.  Medicinal and functional properties;
2.  Consumption for enjoyment;
3.  Supplementary food source;
4.  Negative opinion about the mushroom product range in Hungary.

**Table 5.** Descriptive statistics of items and internal reliability and convergent validity of the latent variables based on the results of Exploratory factor analysis (N = 884).

| Construct | Measurement Item | Mean (SD) | Loadings | Cronbach's Alpha | Composite Reliability (CR) | Average Variance Extracted (AVE) |
|---|---|---|---|---|---|---|
| Medicinal and functional properties | M and F (1) | 5.07 (1.53) | 0.758 | 0.786 | 0.890 | 0.553 |
| | M and F (2) | 4.63 (1.49) | 0.735 | | | |
| | M and F (3) | 5.20 (1.59) | 0.730 | | | |
| | M and F (4) | 4.42 (1.37) | 0.650 | | | |
| | M and F (5) | 5.22 (1.72) | 0.579 | | | |
| | M and F (6) | 4.25 (1.45) | 0.557 | | | |
| Consumption for enjoyment | ENJ1 | 5.06 (1.92) | 0.759 | 0.545 | 0.805 | 0.556 |
| | ENJ2 | 5.91 (1.67) | 0.646 | | | |
| | ENJ3 | 4.94 (1.88) | 0.612 | | | |
| Supplementary food source | SUP1 | 2.77 (1.88) | 0.815 | 0.560 * | 0.865 | 0.641 |
| | SUP2 | 2.33 (1.72) | 0.786 | | | |
| Negative assessment of the mushroom product range | NEG1 | 2.33 (1.72) | 0.798 | 0.526 * | 0.847 | 0.608 |
| | NEG2 | 3.52 (1.73) | 0.762 | | | |

Note: M and F (1) = Mushroom consumption has a beneficial effect on the immune system; M and F (2) = Mushroom consumption has an antitumor effect; M and F (3) = Mushrooms have beneficial health effects; M and F (4) = Mushrooms have a significant vitamin D content; M and F (5) = Mushrooms are an excellent dietary ingredient; M and F (6) = Mushrooms contain antiviral and antibacterial agents; ENJ1 = Taste and aroma of mushrooms; ENJ2 = Mushrooms can be prepared by many methods; ENJ3 = Mushrooms can also be used as a spice; SUP1 = Mushrooms are primarily a meat substitute; SUP2 = Mushrooms are primarily ingredients for special meals; NEG1 = There is a poor choice of mushrooms and mushroom products available in Hungary; NEG2 = There is a poor choice of frozen ready meals with mushrooms in Hungary. * For the two-item constructs, the coefficient Spearman–Brown coefficient was used instead of Cronbach's Alpha.

The construct "Medicinal and functional properties" refers to the high vitamins B, D and C and antioxidant content of mushrooms, as well as their general positive health benefits, such as low calorie content and high quality protein content (which is superior to plant proteins and more similar to animal proteins). Mushrooms also contain antiviral and antibacterial agents. The construct "Consumption for enjoyment" refers to the reliability and safety of farmed mushrooms (as opposed to forest mushrooms). The construct "Supplementary food source" contains consumer attitudes toward and opinions about the range of choices available, the meat substitute properties of mushrooms, and their general role in the diet. The construct "Negative opinion of choices available" refers to the limited range of mushrooms and mushroom products available in Hungary. Based on the results of the path analysis, it was found that the relationships between all dimension pairs are significant, except for the relationship between Supplementary food source and Negative assessment of the mushroom product range (Figure 3). There is a moderate negative correlation (Path Coef. = $-0.52$, $p < 0.001$) between Supplementary food source and Consumption for enjoyment. It can be stated that there is a moderate positive relationship (Path Coef. = 0.47, $p < 0.001$) between Consumption for enjoyment and the Medicinal and functional properties. There is a very weak correlation (Path Coef. = $-0.09$, $p = 0.009$) between Medicinal and functional properties and mushroom consumption as a supplementary food source. Hence, H1 was only partially supported by results: Consumption as a supplementary food source is negatively related to the other two dimensions, which are Medicinal and functional properties and Consumption for enjoyment.

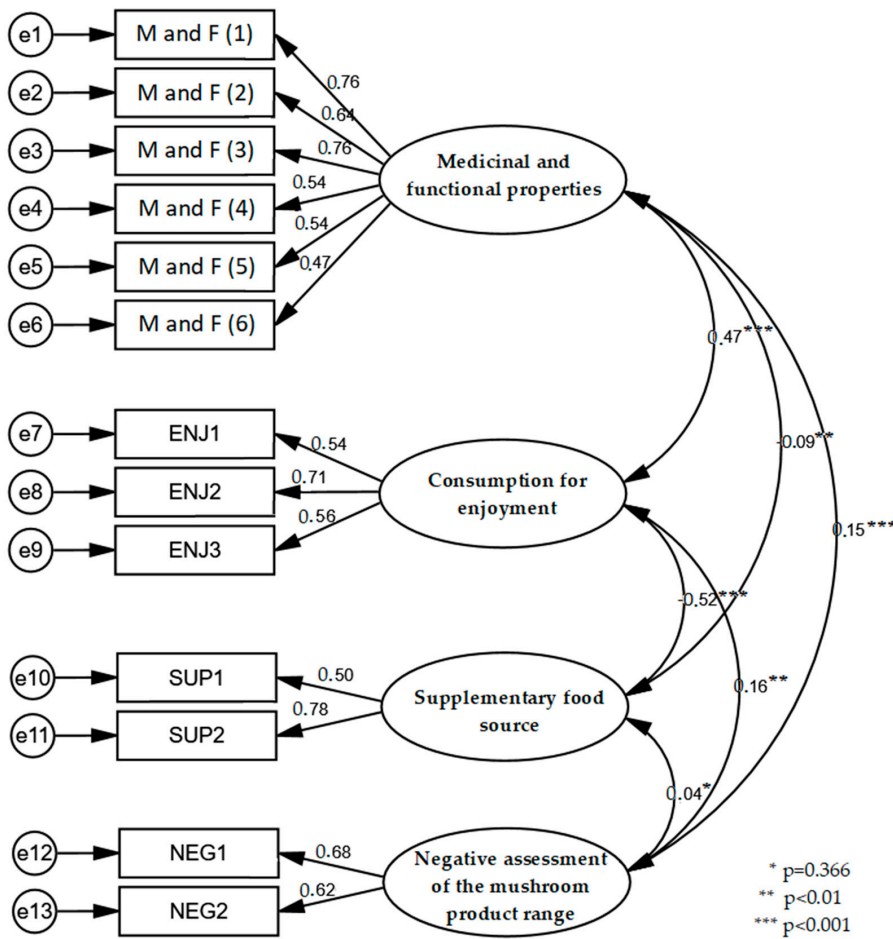

**Figure 3.** Standardized solution for the model with four factors based on Confirmatory factor analysis (N = 884).

The results of the path analysis show that the Negative assessment of the mushroom product range is positively associated with Medicinal and functional properties (Path Coef. = 0.15, *p* < 0.001) and Consumption for enjoyment (Path Coef. = 0.16, *p* < 0.001). However, the positive relationship between the Negative assessment of the mushroom product range and mushroom consumption as a supplementary food source was not significant (Path Coef. = 0.16, *p* < 0.001) and the results only partially confirmed H2.

The abbreviations for the items in the Figure 3 have the following meanings.

M and F (1) = Mushroom consumption has a beneficial effect on the immune system; M and F (2) = Mushroom consumption has an anti-tumor effect; M and F (3) = Mushrooms have beneficial health effects; M and F (4) = Mushrooms have a significant vitamin D content; M and F (5) = Mushrooms are an excellent dietary ingredient; M and F (6) = Mushrooms contain antiviral and antibacterial agents; ENJ1 = Taste and aroma of mushrooms; ENJ2 = Mushrooms can be prepared by many methods; ENJ3 = Mushrooms can also be used as a spice; SUP1 = Mushrooms are primarily a meat substitute; SUP2 = Mushrooms are primarily ingredients for special meals; NEG1 = There is a poor choice of mushrooms and mushroom products available in Hungary; NEG2 = There is a poor choice of frozen ready meals with mushrooms in Hungary.

Observed variables are represented by squares and latent variables are enclosed in circles. Goodness of fit indices include the following: $\chi^2$/df = 4.57; CFI = 0.971; TLI = 0.912; RMSEA (90% CI) = 0.051 (0.046–0.056).

### 3.3. Segmenting Mushroom Consumers

Cluster analysis of Hungarian mushroom consumers was conducted based on opinions about the choice of mushrooms available and consumer consciousness.

Three clusters were identified based on the four dimensions analyzed (Figure 4) and they include the following:

1. Health-conscious consumers;
2. Indifferent consumers;
3. Average consumers.

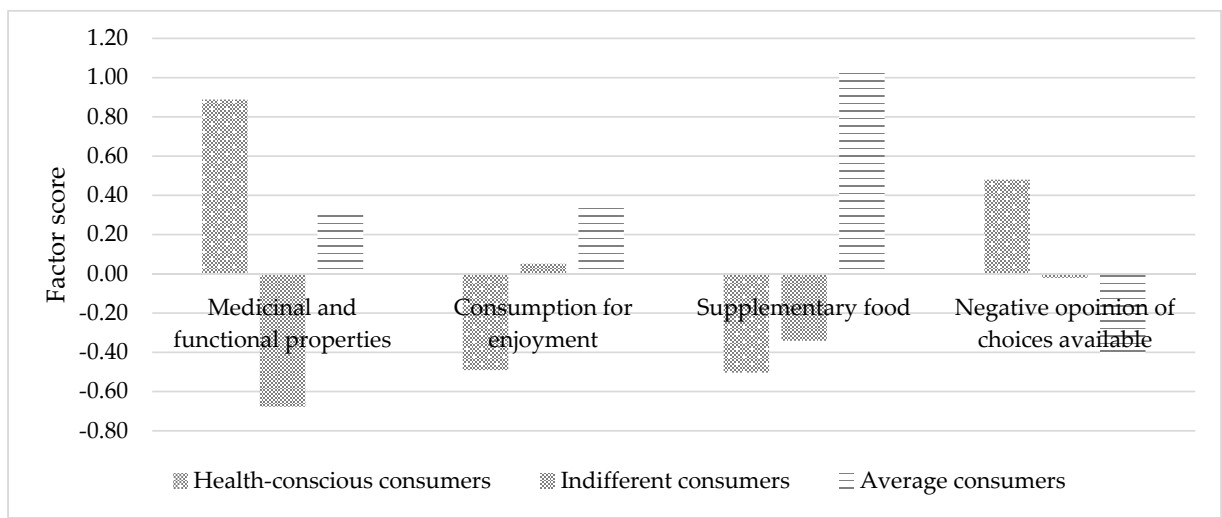

**Figure 4.** Characteristics of the clusters generated based on level of consciousness in consumption.

Health-conscious consumers are most informed about the medicinal and functional properties of mushrooms; at the same time, they have the lowest opinion about the range of products available. On the other hand, the average consumers are not dissatisfied with the choice of mushrooms and are somewhat aware of their health effects and properties and they consider mushrooms a supplementary part of their diet.

Indifferent consumers are unaware of the potential health benefits of mushrooms and are characterized by low consumption for enjoyment (Figure 4).

Based on the results of the Chi-square test (Chi$^2$ = 49.48, $p < 0.001$), a significant correlation can be established between the age group and belonging to the mushroom consumer clusters. Health-conscious mushroom consumers tend to have a higher representation of the 31 to 40 years and 41 to 50 years of age groups than the other two attitude groups, at 19.3% and 25.1%, respectively. Indifferent consumers comprise a high ratio of consumers aged 21 to 30 (42.6%) (Figure 5).

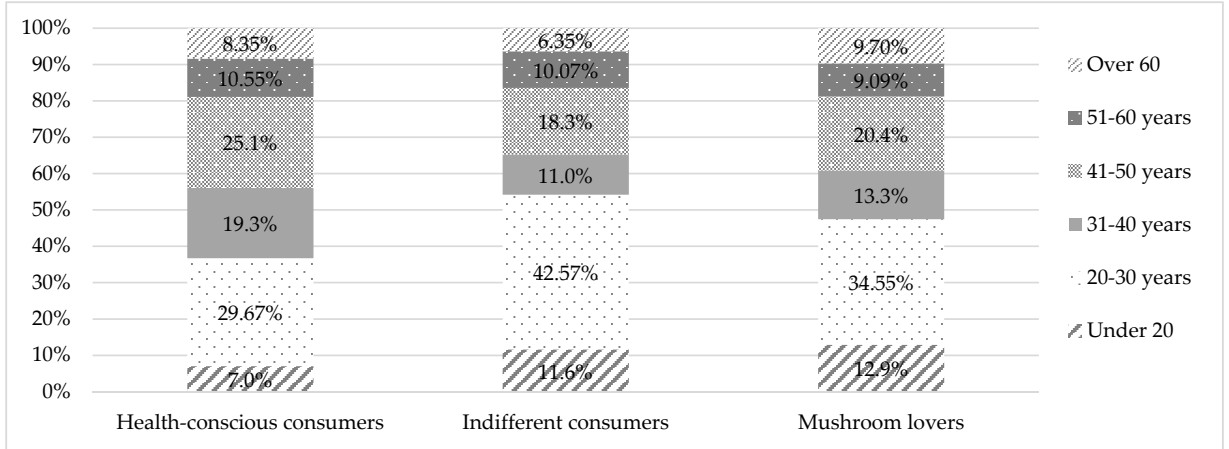

**Figure 5.** Age distribution within clusters generated based on the level of consciousness in mushroom consumption.

The result of the Chi-square test (Chi$^2$ = 20.32, $p$ = 0.002) indicates a significant relationship between education and belonging to mushroom consumer clusters. Health-conscious mushroom consumers have a lower representation of secondary school graduates than the other groups (47.3%). The proportion of higher education graduates is significantly higher (20.9%) in this cluster than in the case of indifferent consumers and average consumers (Figure 6).

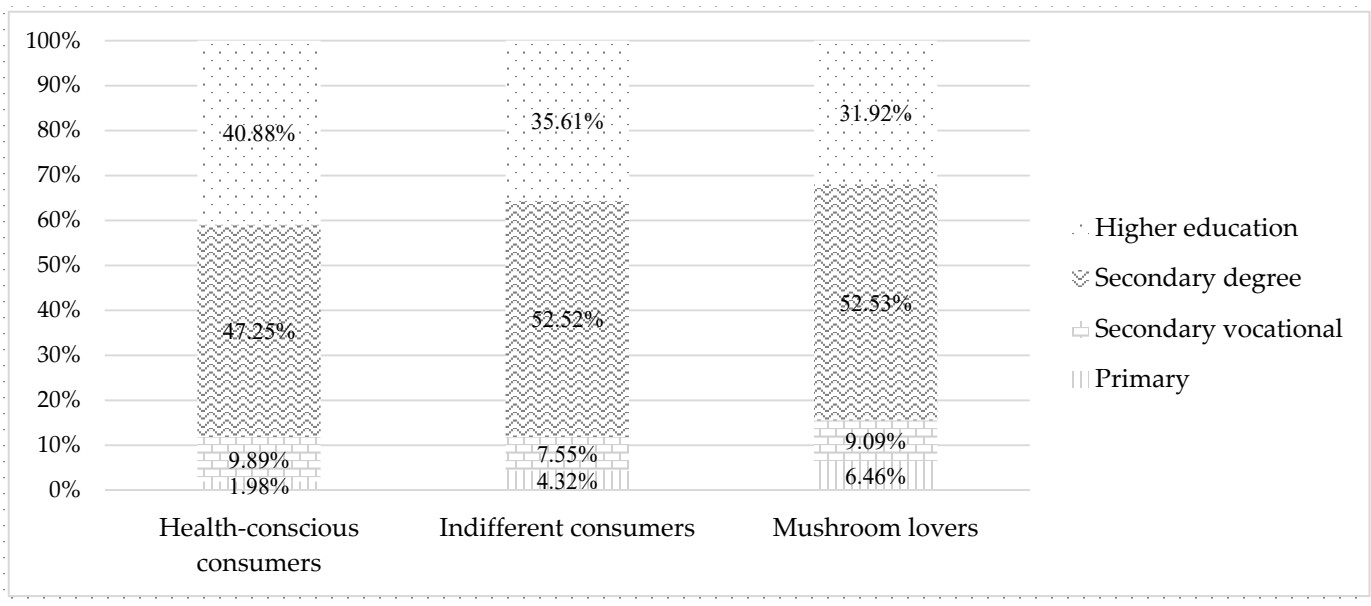

**Figure 6.** Distribution by educational level within clusters generated based on level of consciousness in mushroom consumption.

The result of the Chi-square test (Chi$^2$ = 21.60, $p$ < 0.001) confirms a significant correlation between marital status and belonging to mushroom consumer clusters. Health-conscious mushroom consumers tend to be married or in a partnership (59.3%). Indifferent mushroom consumers have a higher proportion of single people than compared to the other attitude groups (43.9%) (Figure 7).

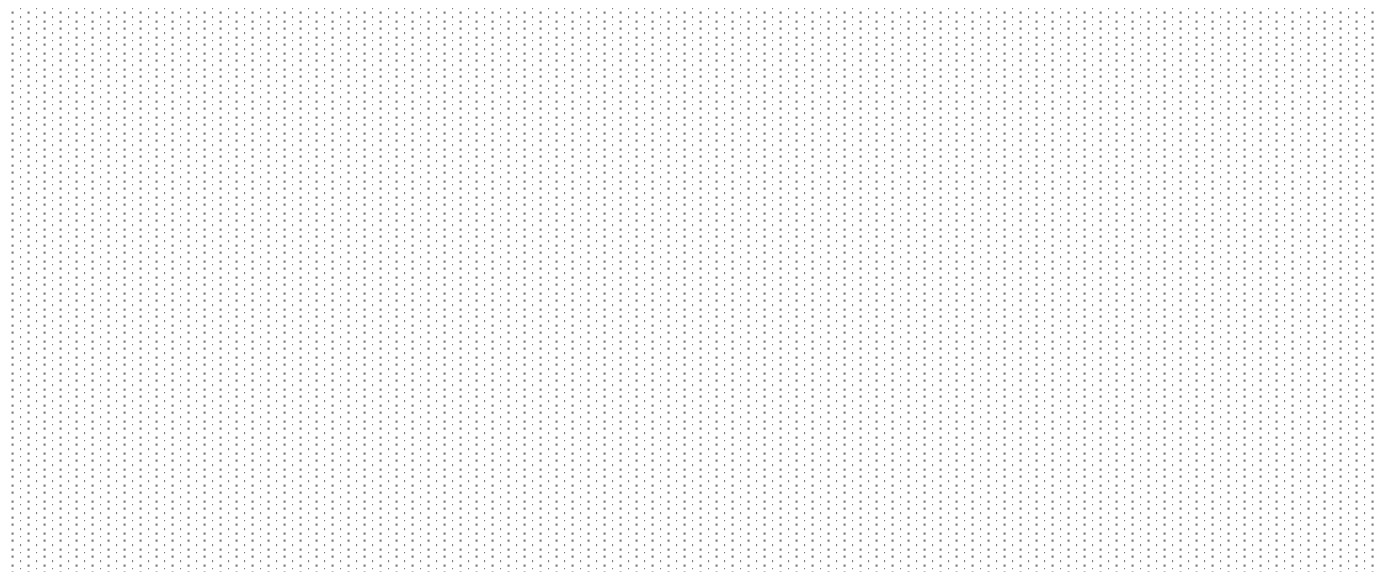

**Figure 7.** Distribution of family status within clusters generated based on level of consciousness in mushroom consumption.

Location does not seem to show significant differences among the three clusters (Chi$^2$ = 13.07, $p$ = 0.109). Indifferent consumers comprise mostly respondents living in the

capital city (34.9%), but also have a high representation consisting of town dwellers (34.4%). Health-conscious consumers and average consumers have a similar distribution (Figure 8).

**Figure 8.** Distribution by location within clusters generated based on level of consciousness in mushroom consumption.

In support of H3 the results confirm the impact of consumers' age, education, and marital status on their mushroom consumer attitudes.

## 4. Discussion

Several research related to food consumption habits [87–91,105] justified that consumer behavior differs by gender and other socio-demographic characteristics of respondents. The authors' results are confirmed these findings.

Based on the survey data obtained for Hungarian consumers, four dimensions of mushroom consumer habits were identified and are included in several food consumer models.

The research results do not prove that urban dwellers are more health-conscious about mushroom consumption. Among health-conscious consumers, the proportion of single people is significantly lower, which can be explained by the fact that single people are also less health-conscious in general than couples [106] and mainly couples with children.

Those with higher education represent a higher proportion of health-conscious mushroom consumers. In younger age groups (30 years and younger), the proportion of health-conscious mushroom consumers is lower. For the "Medicinal and functional properties dimension", Wang's Healthy Dietary Behavioral Model [79] was utilized. "Consumption for enjoyment" is related to Ajzen and Fishbein's model [92] regarding consumer beliefs and other background variables. According to the authors' results, the moderate negative correlation between Supplementary food source and the Consumption for enjoyment is probably because people who use mushrooms as a garnishing tend to follow national cuisine and do not use mushrooms as a spice [107]. The moderate positive relationship between Consumption for enjoyment and Medicinal and functional properties can also be attributed to the fact that knowledge of the healing effects of mushrooms is intertwined with a greater knowledge of species, thus making them more aware of the enjoyment value of mushrooms. The authors' findings are in line with the results of some of the latest studies [19,88,108] that revealed the relationship among the reasons for mushroom consumption.

The negative assessment of the Hungarian mushroom selection shows a weak positive correlation with the Medicinal and functional properties dimension, which can be explained by the fact that the market for medicinal mushrooms is indeed very underdeveloped in Hungary.

A consumer survey also confirms a weak positive correlation between the negative assessment of the Hungarian mushroom selection and mushroom consumption for enjoy-

ment. This finding can be explained by the fact that "mushroom lovers" are more critical of the mushroom product range.

## 5. Conclusions

There is no doubt that sustainability oriented consumption is a megatrend that influences the consumer habits of mainly older generations and family with children.

Mushrooms are still controversial: Even though their positive health effects are undeniable, their consumption varies by cultures and countries. Mushroom consumption in Hungary lags behind the global average. Factors describing mushroom consumption were arranged into four major groups and three groups are related to the reasons for consuming mushrooms. The results were used to segment Hungarian mushroom consumers, which could provide valuable insights for producers and traders to adjust their production and marketing strategies to the needs and expectations of various consumer segments.

The majority of consumers consider mushrooms a supplementary food source, which coincides with local traditions. It is significant that mushrooms having medicinal and functional properties is the second most important consideration and this is due to active marketing by some producers and chambers of commerce; however, that also shows that the increasing share of older population and families with higher education possess positive mindsets. Future marketing activities must target further populations and spread the recognition of mushrooms as an important food source.

It was somewhat surprising that the relatively low per capita consumption of mushrooms is not due to a general negative opinion of or aversion to mushrooms. The mushroom sector is facing the following challenges in achieving its goal of increasing consumption:

- A specific marketing strategy is needed and this can be conducted by relying on experience from models in other countries targeted at the youngest and oldest generations (population under 20 and over 65 years of age);
- It would be beneficial to create marketing strategies tailored to the separate mushroom consumer segments established in the current study and other similar studies in the future;
- Educational programs, such as the School Mushroom program by Bio-Fungi Ltd., Ócsa, Hungary need to be adapted to all levels of education from preschool to secondary in a nationwide campaign;
- Intensive marketing campaigns in the future need to highlight the potential health benefits of mushrooms in the diet;
- Transfer of knowledge needs to be strengthened by relying on formal education as well as social media to enhance positive attitudes and habits and to establish a healthy new generation in the future.

## 6. Limitations of the Study and Future Directions

In the present research, we have not paid special attention to the Z and Alpha generations. They are the customers and parents of the future and a wide range of influences and experiences shape their consumer habits. Exploring these generations is a research task for the near future. In our research, we also want to deepen the in-depth analysis of the impact of environmental factors on protein consumer behavior and to look for the best methods to transfer new knowledge, especially for the age groups most affected by the digital environment. Future studies could also examine the role of environmental concerns and lifestyle aspects in determining Hungarian consumers' eating habits towards mushrooms and mushroom products.

**Author Contributions:** B.B., M.F.-F. and S.V. performed research idea conceptualization; B.B. and S.V. collected the research data; B.B. and S.V. designed the research methodology and performed the formal analysis and initial drafting of the results; review and editing, M.F.-F. All authors have read and agreed to the published version of the manuscript.

**Funding:** This research received no external funding.

**Institutional Review Board Statement:** Not applicable.

**Informed Consent Statement:** Not applicable.

**Data Availability Statement:** The data presented in this study are available upon request from the corresponding author.

**Conflicts of Interest:** The authors declare no conflict of interest.

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
