# Peer review of "An Analysis of Mushroom Consumption in Hungary in the International Context"

_agriculture, doi:10.3390/agriculture11070677_

Round 1
Reviewer 1 Report
Dear authors,
Thank you for the possibility to read and evaluate your paper.
I am sending you my feedback in terms of Originality, Relationship to Literature, Methodology, Results, and Implications for research, practice and/or society.
My overall evaluation of the paper is a minor revision.
1. Originality
The theme of the article is actual and it brings new information on mushroom consumption in Hungary in comparison with other countries. The aim of the article is well developed and corresponds with the article title.
2. Relationship to Literature
The relationship to literature is also well developed, but more papers from last years (covering 2019–2021) should be included. At the end of the Literature review part, I miss a formulation of research hypotheses that would be connected with the previously published studies.
3. Methods
The Methods part is clear. I appreciate a huge sample of respondents in this study but more information should be devoted to the sampling technique. When presenting items from the questionnaire and the EFA, I would recommend a paragraph on the common method bias and how has it influence the results of the study.
4. Results
The Results part is clear. The discussion part should be more detailed and discuss better results of the study compared to the previous studies.
5. Implications for research, practice and/or society
Implications for research, practice and/or society are well written.
Author Response
We would like to thank you for the thorough comments given for our article. We have diligently reviewed all the raised comments and concerns and have taken actions to correct and improve this paper. We have made changes to the text using the "Track Changes" feature in Microsoft Word.
- Originality
The theme of the article is actual and it brings new information on mushroom consumption in Hungary in comparison with other countries. The aim of the article is well developed and corresponds with the article title.
Thank you very much for your positive evaluation.
- Relationship to Literature
The relationship to literature is also well developed, but more papers from last years (covering 2019–2021) should be included. At the end of the Literature review part, I miss a formulation of research hypotheses that would be connected with the previously published studies.
Thank you for rising our attention for updating the reference list, some new citations are added and we improved the literature review using them These help us in the formulation of Hypotheses and we added them to the text in a separate paragraph. It also helped us to develop the discussion part.
- Methods
The Methods part is clear. I appreciate a huge sample of respondents in this study but more information should be devoted to the sampling technique. When presenting items from the questionnaire and the EFA, I would recommend a paragraph on the common method bias and how has it influence the results of the study.
Thank you for highlighting this issue. The results of examining potential common method variance can be found in L342-348.
- Results
The Results part is clear. The discussion part should be more detailed and discuss better results of the study compared to the previous studies.
The discussion part has been improved taken also into consideration the results of more recent related research results. Thank you for the suggestion.
- Implications for research, practice and/or society
Implications for research, practice and/or society are well written.
Thank you for your positive feedback.

Reviewer 2 Report
Review of the manuscript: An Analysis of Mushroom Consumption in Hungary in International Context.
KEYWORDS: I strongly suggest authors to introduce more keywords. The usefulness of keywords is to make the article both more and more easily searchable visible after its publication through commonly used search engines.
Introduction: The introduction is interesting, but in my opinion, it does not fully cover the topic. Below are some suggestions on how to expand this section. Moreover, the introduction is based on only 7 literature items. I propose to develop the introduction based on more current literature.
Literature Review: Note that, out of all cited items, some are older than 10 years. The authors refer to some very old literature. Can these items not be replaced with newer ones? I understand that it is the significance of mushroom production and consumption in a historical view, but this type of work published in a journal with high impact factors should be based on the latest and up-to-date knowledge.
I believe that the work presented for review is of a high technical level, but it requires substantive amendments (please post new items).
I am asking for a deeper description, taking into account my suggestions above, with post new items.
Author Response
We would like to thank you for the thorough comments given for our article. We have diligently reviewed all the raised comments and concerns and have taken actions to correct and improve this paper. We have made changes to the text using the "Track Changes" feature in Microsoft Word.
KEYWORDS: I strongly suggest authors to introduce more keywords. The usefulness of keywords is to make the article both more and more easily searchable visible after its publication through commonly used search engines.
Thank you for your suggestion, We have introduced two more keywords (functional food, SEM).
Introduction: The introduction is interesting, but in my opinion, it does not fully cover the topic. Below are some suggestions on how to expand this section. Moreover, the introduction is based on only 7 literature items. I propose to develop the introduction based on more current literature.
Thank you, the number of recent references has been increased in the introduction part.
Literature Review: Note that, out of all cited items, some are older than 10 years. The authors refer to some very old literature. Can these items not be replaced with newer ones? I understand that it is the significance of mushroom production and consumption in a historical view, but this type of work published in a journal with high impact factors should be based on the latest and up-to-date knowledge.
Thank you for rising our attention We found fresh related literatures, we have reviewed and cited them accordingly in the revised manuscript, both in the literature and introduction part.
I believe that the work presented for review is of a high technical level, but it requires substantive amendments (please post new items). I am asking for a deeper description, taking into account my suggestions above, with post new items.
Thank you for your positive feedback and suggestion, based on your suggestion the manuscript has been substantially revised. some new items have been added and evaluated. Thank you for your valuable contribution to the improvement of our paper.
